# Personality variation is eroded by simple social behaviours in collective foragers

**Sean A. Rands** *, **Christos C. Ioannou**

School of Biological Sciences, University of Bristol, United Kingdom

* sean.rands@bristol.ac.uk

## Abstract

The movement of groups can be heavily influenced by 'leader' individuals who differ from the others in some way. A major source of differences between individuals is the repeatability and consistency of their behaviour, commonly considered as their 'personality', which can influence both position within a group as well as the tendency to lead. However, links between personality and behaviour may also depend upon the immediate social environment of the individual; individuals who behave consistently in one way when alone may not express the same behaviour socially, when they may be conforming with the behaviour of others. Experimental evidence shows that personality differences can be eroded in social situations, but there is currently a lack of theory to identify the conditions where we would expect personality to be suppressed. Here, we develop a simple individual-based framework considering a small group of individuals with differing tendencies to perform risky behaviours when travelling away from a safe home site towards a foraging site, and compare the group behaviours when the individuals follow differing rules for aggregation behaviour determining how much attention they pay to the actions of their fellow group-members. We find that if individuals pay attention to the other members of the group, the group will tend to remain at the safe site for longer, but then travel faster towards the foraging site. This demonstrates that simple social behaviours can result in the repression of consistent inter-individual differences in behaviour, giving the first theoretical consideration of the social mechanisms behind personality suppression.

## Author summary

The movement of groups can be heavily influenced by 'leader' individuals who differ from the others in some way. A major source of differences between individuals is the repeatability and consistency of their behaviour, commonly considered as their 'personality', which can influence both their position within a group as well as their tendency to lead. However, links between personality and behaviour may also depend upon the immediate social environment of the individual; individuals who behave consistently in one way when alone may not express the same behaviour socially, when they may be conforming with the behaviour of others. Experimental evidence shows that personality differences can be eroded in social situations, but there is currently a lack of theory to identify the

**Data Availability Statement:** Data and code used for generating the figures are freely available on Figshare at https://doi.org/10.6084/m9.figshare.

21996242. All other data are within the manuscript and its supplementary information.

**Funding:** SAR was supported by the University of Bristol Returning Carers' Scheme, CCI was supported by a Natural Environment Research Council (NERC) Independent Research Fellowship (NE/K009370/1), and both authors were supported by a NERC standard grant (NE/P012639/1). The funders had no role in study design, data collection and analysis, decision to publish, or preparation of the manuscript.

**Competing interests:** No competing interests are declared.

conditions where we would expect personality to be suppressed. Here, we develop a simple model of a small group of individuals who differ in their tendency to perform risky behaviours when travelling away from a safe home site towards a foraging site. We find that if individuals pay attention to the other group members, this has an overall impact on the efficiency of the group, and demonstrates that simple social behaviours can result in the suppression of individual personalities.

## Introduction

How groups move collectively can depend upon which individuals have influence [1–3]. Much attention has been paid to the behaviour of groups when individuals can behave autonomously but also respond to the actions of their neighbours, and models have been successful at simulating group behaviour that corresponds to natural patterns of flocking and shoaling. These models typically consider individuals making movement decisions based on the presence or absence of local neighbours and their proximity [4–11], where individuals in close proximity may copy, or be strongly influenced by, each other's behaviour (*e.g.* [12–15]). The behaviour of individuals in a group can be disproportionately influenced by a few individuals, who act as leaders [16–19]. These individuals may hold useful information [20–23] and influence the movement of the group solely through their actions [24–27], or through the imposition of social hierarchies [28,29]. Physical differences between individuals may cause some individuals to take specific positions within a group [30,31]. Leadership decisions may also be made by individuals who are in a specific behavioural or physical state at a given moment in time [32–35]. Consistent with this, behavioural and physiological state can influence the position of individuals within groups where hungry individuals position themselves at the front of the group where they are more likely to find food [36–38], trading an increased chance of energetic gain for an increased risk of predation [39]. The movement of individual fish in shoals is influenced by their nutritional state, and it is likely that the behaviours of single fish is dependent not only on their own state, but also that of their neighbours [40].

These mechanisms demonstrate that the movement of groups can be heavily influenced by individuals who differ in some way, regardless of whether this inter-individual variation is long-term or transient. Apart from physical differences such as size, a major source of long-term differences between individuals is in the repeatability and consistency of their behaviour. This is commonly considered to be the 'personality' of an individual animal [41–43]. The personalities of individuals can influence both their position within a group [44–46], and their tendency to lead [47,48]. For example, consistently 'bold' individuals are willing to accept greater risk in the same manner that leaders in many animal groups do, and hence bold individuals often lead group decisions [48,49]. However, even if individuals are behaviourally consistent over time and contexts, their behaviour may still be influenced by their social environment [50]. Individuals who behave consistently on their own may not express this behaviour in a group when they conform to other group members' behaviour [45,51], and this conformity effect can be stronger in more sociable individuals who are more likely to be influenced by others' behaviour [52].

While these experimental results show that strong personality differences can be eroded in social situations, there is a lack of modelling to establish under what conditions we would expect inter-individual (*i.e.* personality) variation to be suppressed, or still be expressed, when individuals interact in groups. The study by McDonald *et al.* [45] considered the group decisions made by small fish (three-spined sticklebacks, *Gasterosteus aculeatus*) choosing to cross

an open space between a refuge and a known food source. These fish could be seen as making a typical trade-off between perceived predation risk (elevated when crossing the open space and at the foraging site) and starvation [53–55]. When travelling this short distance on their own, the behaviour of solitary fish reflects this risk-taking trade-off [56] and was shown to be repeatable, *i.e.* the fish demonstrated personality variation in boldness [45]. However, when in groups, the fish were influenced by social factors as well: being in a group reduced the time taken to travel between the refuge and the food source for the average individual, but it also suppressed personality variation. Here, we present a model motivated by the experiment conducted by McDonald *et al.* [45] that considers what happens when we create groups from individuals who would be consistently different if they were behaving on their own. Assuming that there are benefits for reducing predation risk from remaining in close proximity to other group members, we also explore the effects of simple social interaction rules that may influence the behaviour of individuals, echoing previous models considering social herding behaviour under predation risk [57–61]. Using this model, we explore whether simple social interaction rules are sufficient to erode consistent behavioural differences between individuals.

## Methods

### The models

The models consider a linear (one dimensional) environment, consisting of a home refuge site, a foraging site located in the region greater or equal to $d_{food}$ distance units away from the home site in one direction, and a continuous strip of food-free space between them (sketched in Fig 1). Within the environment, there are $n$ individuals, and at a given time period $t$ each individual $i$ is located at a point $d_{i,t}$ distance units away from the home site. All individuals are assumed to have constant personal speeds, so if an individual moves during a period, it moves $s_i$ distance units towards or away from the home site. Whether and where to move is dependent upon the individual's probability of moving outward to the foraging site, which is characterised by two variables. The first of these is a time-dependent probability of moving outward $p_{i,t}$, which increases as time passes and represents a growing need by an individual to move to the site where food is available. Secondly, individuals are characterised by a personal outward probability adjustment $\omega_i$, which represents the boldness of the individual. If the model includes social behaviour, the decision is also influenced by the current position of the other individuals within the environment.

At the start of a simulation $t = 0$, $n$ individuals are created. Each individual $i = 1 \ldots n$ has a time-dependent probability of moving outward $p_{i,t}$, which is initially set at a baseline common to all individuals, $p_{i,0} = p_{baseline}$. At the same time, the individual's personal outward probability adjustment is set at $\omega_i = (i-1) \cdot \omega_{difference}$. Individual 1 has a personal outward probability adjustment set at 0, and the other individuals have personal outward probability adjustment values that increase regularly by $\omega_{difference}$. This means that the identity $i$ of an individual defines its likelihood of choosing to move outwards, falling on a movement hierarchy where individuals with low values of $i$ are less likely to move. The speed $s_i$ of individual $i$ is independently sampled from a uniform distribution $U(0.95, 1.0)$. The initial location of all individuals is at the home site, so $d_{i,0} = 0$. Table 1 sketches out how individual parameters are fixed (or change) over the course of a simulation.

Subsequent periods are then considered sequentially, starting at $t = 1$. During a period, each individual makes a single decision (sketched in Fig 1), and the order that the different individuals make and then act on their decision is randomised at the start of each period. Firstly, the

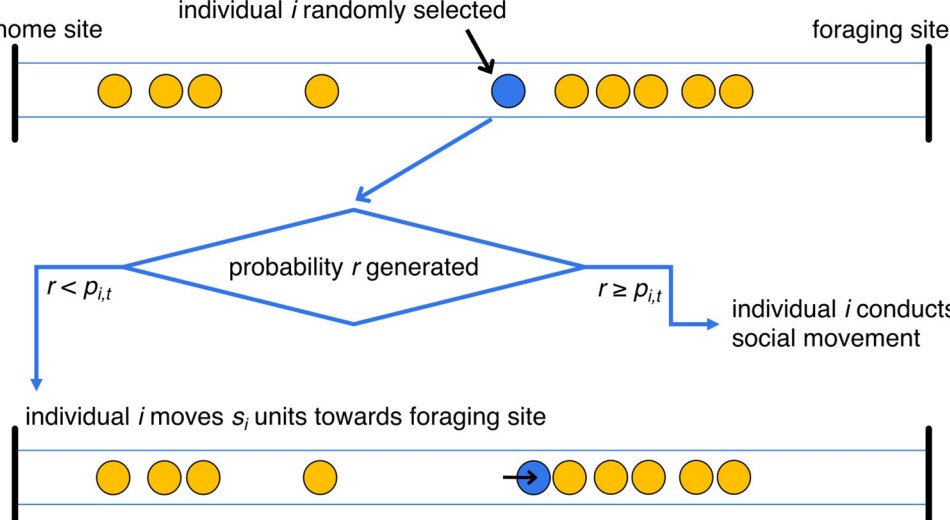

**Fig 1. Sketch of the movement decision-generating process.** The figure shows an individual who has not yet reached the foraging site, at some intermediate timestep during a simulation. Within a timestep, all individuals make a single movement decision, and the order in which they do this is randomised. If an individual chooses to conduct a social movement, it then moves according to one of the rules sketched in Fig 2.

time-dependent probability of moving outward is adjusted to

$$p_{i,t} = p_{i,t-1} + \omega_i \qquad (1)$$

(so at each timestep, each individual becomes more likely to move outward, and the difference between individuals in their values of $\omega_i$ means that outward movement probability increases faster for bolder individuals, assuming that all individuals have the same knowledge of the foraging site). The individual then makes its movement decision: if it is at the foraging site ($d_{i,t-1} \geq d_{food}$), it does not move (so $d_{i,t} = d_{i,t-1}$); if the refuge site lies between itself and the foraging site (so $d_{i,t-1} < 0$, which computationally could occur at the beginning of a simulation due to small floating-point errors in the calculation), it moves towards to foraging site such that $d_{i,t} = d_{i,t-1} + s_i$; whereas if it is positioned between the refuge and foraging site ($0 \leq d_{i,t-1} < d_{food}$), it draws a random value from a uniform distribution $U(0,1)$. If this random value is

**Table 1. Changes in an individual's variables over the course of a simulation.**

| individual, $i$ | speed, $s_i$ | personal outward probability adjustment, $\omega_i$ | time-dependent probability of moving outward, $p_{i,t}$ | | | | |
|---|---|---|---|---|---|---|---|
| | | **fixed at beginning of simulation** | **altered at each timestep** | | | | |
| | | | $t = 0$ | $t = 1$ | $t = 2$ | $t = 3$ | ... |
| 1 | $s_1$ | $\omega_1 = 0$ | $p_{1,0} = p_{baseline}$ | $p_{1,1} = p_{baseline} (= p_{baseline} + \omega_1)$ | $p_{1,2} = p_{baseline}$ | $p_{1,3} = p_{baseline}$ | ... |
| 2 | $s_2$ | $\omega_2 = \omega_{difference}$ | $p_{2,0} = p_{baseline}$ | $p_{2,1} = p_{baseline} + \omega_2$ | $p_{2,2} = p_{baseline} + 2 \cdot \omega_2$ | $p_{2,3} = p_{baseline} + 3 \cdot \omega_2$ | ... |
| 3 | $s_3$ | $\omega_3 = 2 \cdot \omega_{difference}$ | $p_{3,0} = p_{baseline}$ | $p_{3,1} = p_{baseline} + \omega_3$ | $p_{3,2} = p_{baseline} + 2 \cdot \omega_3$ | $p_{3,3} = p_{baseline} + 3 \cdot \omega_3$ | ... |
| 4 | $s_4$ | $\omega_4 = 3 \cdot \omega_{difference}$ | $p_{4,0} = p_{baseline}$ | $p_{4,1} = p_{baseline} + \omega_4$ | $p_{4,2} = p_{baseline} + 2 \cdot \omega_4$ | $p_{4,3} = p_{baseline} + 3 \cdot \omega_4$ | ... |
| ⋮ | ⋮ | ⋮ | ⋮ | ⋮ | ⋮ | ⋮ | |

### 1. CENTRAL rule

i. focal (blue) individual identifies **mean centre** of other (orange) individuals

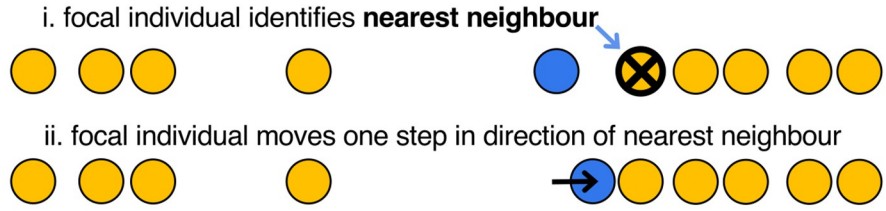

ii. focal individual moves one step in direction of mean centre

### 2. NEAREST NEIGHBOUR rule

i. focal individual identifies **nearest neighbour**

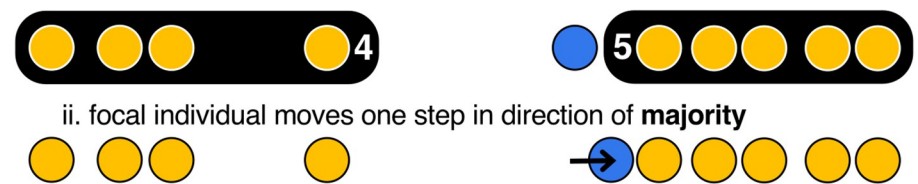

ii. focal individual moves one step in direction of nearest neighbour

### 3. MAJORITY rule

i. focal individual compares numbers of individuals on either side of itself

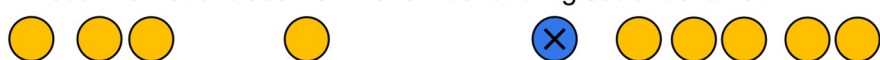

ii. focal individual moves one step in direction of **majority**

### 4. NON-SOCIAL rule

focal individual **does not move** if conducting social behaviour

**Fig 2. Sketches of the four social interaction rules.** Each simulation only considers a single rule, which all individuals follow when they choose to socially interact.

below $p_{i,t}$, it moves towards the foraging site ($d_{i,t} = d_{i,t-1} + s_i$), otherwise it conducts a social action dependent upon which of the four social rules it uses.

The social interaction rules all calculate a metric that allows the individual to choose a direction of travel (as sketched in Fig 2). They are:

1. **central**: move towards the mean centre of group. For individual $i$, the mean centre of the group is calculated as the mean location of the other group members, so the calculated centre occurs at $d' = (\sum_{\forall j \neq i} d_j)/(n-1)$. Having calculated $d'$, the individual then moves $s_i$ units in the direction towards (or past) $d'$ (or does not move if its current position is already exactly $d'$).

2. **nearest-neighbour**: move towards the nearest neighbour. The focal individual identifies its closest neighbour (or randomly chooses a nearest neighbour, if there are more than one),

and then moves $s_i$ units in the direction towards (or past) the identified neighbour. If the nearest neighbour occupies $d_{i,t-1}$ the focal individual does not move.

3. **majority**: move towards majority of group. The focal individual identifies whether there are more colleagues towards the foraging site or the refuge. It then moves $s_i$ units in the direction of the majority. If there are equal numbers of individuals in either direction, the focal individual does not move. Although similar to the central rule above, this movement rule is not influenced by large separations that might occur with the slowest and fastest group members.

4. **non-social** behaviour. Here, the focal individual does not move if the 'social' behaviour is chosen.

All individuals act on a single movement decision once during a period. A simulation continues until all individuals are within the foraging site at the end of a period.

## Model exploration: Effects of model parameters

Two parameters were identified that could affect the behaviour of the modelled group: the baseline time-dependent probability of moving outward $p_{baseline}$ (where an increasing value means that individuals are going to start moving outwards after a shorter period of time, *i.e.* are bolder), and the outward probability adjustment parameter $\omega_{difference}$ (where a larger value means that there is an increase in the separation in probabilities that individual group members move outward, *i.e.* they become more different from one another). The distance to the foraging site $d_{food}$ and the size of the group $n$ were also assumed to affect group behaviour. Using baselines of $\omega_{difference}$ = 0.001, $p_{baseline}$ = 0.001, $d_{food}$ = 100 units and $n$ = 10, we explored the effects of altering these target parameters by systematically changing a single parameter away from its baseline and running 10,000 simulations for each resulting parameter set for each of the four social interaction rules (including the non-social behaviour). These single parameter explorations considered $p_{baseline}$ = (0.0001, 0.0002, . . . 0.0020), $\omega_{difference}$ = (0.0001, 0.0002, . . . 0.0020), $d_{food}$ = (20, 30, . . . 200 units), and $n$ = (5, 10, . . . 50). Simulations were built and run using *NetLogo* 5.3.1–6.2 [62,63], and example code is available for use [64].

Six statistics were recorded from each simulation to measure group behaviour. The number of periods until the end of each simulation was recorded to measure the number of timesteps taken for all group members within the simulation to reach the foraging site. One individual from each simulation was also chosen at random to be a focal individual to collect statistics from. As well as the identity of this individual (corresponding to its position within a movement hierarchy), we recorded the time it took to first move past a threshold distance (taken to be ten distance units from the refuge, referred to as its 'departure' time) and the latency between this individual's time and that of the first individual to pass this threshold within the simulation. We also recorded the time it took the focal individual to reach the foraging site, and the latency between this time and that of the first individual to reach the foraging site within the simulation. Because each individual in a group was equally likely to be the randomly chosen focal individual, recorded latency values therefore represent an average latency after the first individual to pass the threshold or arrive at the foraging site (where if the first individual to arrive is being tracked, it is assumed to have a latency of zero). From these statistics, we also calculated the time each focal individual took to travel between crossing the threshold and arriving at the foraging site.

Data were explored within *R* 3.3.0–4.1.2 [65]. Exploratory graphs were generated using *ggplot2* 3.3.5 [66] both for the average of a simulated group (calculated as the mean value for all individuals followed for a given parameter value, regardless of the hierarchical identity of

those individuals), and for focal individuals with a known position within a movement hierarchy (although the latter was not considered when changing $n$, due to the confounding effects of increasing the size of the group). In order to quantify the difference in variation within and between the increments of target parameters during the exploration, we ran standard ANOVA tests with the altered parameter as the explanatory variable and the behavioural measure as the response. We extracted the resulting $F$ values as a measure of the ratio of variance between and within each of the increments of the target parameter. Larger $F$ values indicate larger effects of changing the target parameter relative to the variation between replicates of the simulation with identical parameters.

## Model exploration: Variation between individuals

We also ran smaller sets of simulations where the travel time, departure time and arrival time of all individuals within a group of 10 individuals were recorded. This allowed quantification of how difficult it was to identify differences between interacting individuals within a simulation based on the inter-individual variation in their personal outward probability adjustment $\omega_i$ (equivalent to personality variation in boldness). We tested how the variation between individuals within a simulation was shaped by the four parameters ($\omega_{difference}$, $p_{baseline}$, $d_{food}$ and $n$) considered in the models. To do this, we ran four sets of simulations where three of the parameters were held constant, and the fourth was systematically altered. Using baselines identical to the earlier model ($\omega_{difference}$ = 0.001, $p_{baseline}$ = 0.001, $d_{food}$ = 100 units, $n$ = 10), we systematically changed a single parameter away from its baseline, using $p_{baseline}$ = (0.0001, 0.0002, . . . 0.0020), $\omega_{difference}$ = (0.0001, 0.0002, . . . 0.0020), $d_{food}$ = (20, 30, . . . 200 units), or $n$ = (5, 10, . . . 50), and ran 200 simulations for each resulting parameter set for each of the four social interaction rules (including the non-social behaviour). This systematic alteration of a single parameter meant that we were able to generate four datasets where there was a large and evenly-spread amount of variation in a single parameter, with no variation in the other three parameters. This meant that we could observe how individual differences in behaviours within a group could be affected if there was variation in a single parameter.

The travel time, departure time and arrival time for all individuals in the set were rescaled such that the minimum time within the group was set equal to 0 and the maximum to 1. In order to quantify how difficult it was to distinguish individuals within a group, we measured the variation between individuals within each group (*i.e.* within each run of the simulation) relative to the variation between groups (different runs of the simulation with the same set of parameters). This was done using repeated-measures ANOVA tests with the value of the single altered model parameter and the identity of the individuals (their position within the initial movement hierarchy) repeated within a simulation as the two explanatory variables, and the behavioural measure as the response (the rescaled timings for each individual), and extracted the resulting $F$ values. Larger $F$ values indicate greater between-group variation, so that individuals within the same group show more similar behaviour, and hence inter-individual variation within a group is more difficult to detect.

To explore the difference between social and non-social behaviours, for the simulation sets described above where $\omega_{difference}$, $p_{baseline}$, $d_{food}$ and $n$ were systematically altered we calculated the difference between the rescaled time parameters for each of the three social interaction rules and the non-social behaviour, giving a value between -1 (the focal individual took the minimum time when behaving socially and the maximum time when behaving non-socially) and +1 (the focal individual took the maximum time when behaving socially and the minimum time value when behaving non-socially).

## Results

In most cases and as expected, increasing either of the parameters generating outward movement ($\omega_{difference}$ and $p_{baseline}$) led to all measured behaviours happening faster or sooner (Figs 3 and S1, where the transition from green to yellow indicates an increase the value of the target parameter), regardless of the social interaction rule being used. Similarly, increasing the distance to the foraging site tended to cause behaviours to take longer or happen later (S2 Fig), whilst increasing group size meant behaviours tended to occur sooner or faster (S3 Fig).

For most of the parameter sets investigated, the groups with social interactions conducted the measured behaviours faster or sooner than the corresponding groups of independently-behaving non-social individuals (Figs 3 and S1–S3). Confirming that the central and majority rules resulted in greater cohesion of the groups, the latencies between the first and other individuals to leave the refuge and to arrive at the foraging site (Fig 3J–3L and 3P–3R, and panels J-L and P-R in S1–S3 Figs) were much faster with these rules compared to the nearest neighbour and non-social rules, which were more similar to one another.

Direct pairwise comparisons between the social rules (S4–S7 Figs) demonstrate that the different social interaction rules have differing effects upon the behaviours observed. The total time until the end of the simulation was minimised by the central and (to a lesser extent) majority rules (demonstrated in the top panels on S4–S7 Figs, where the panels for the central and majority rules showed very negative values for most of the parameter space explored, meaning that individuals following either the central or majority rule took much less time to finish the simulation than the strategy they are compared against), and took the longest in the simulations with no social interactions between the individuals (*e.g.* the top left panel of S4 Fig shows that non-social groups finished between ~140 and ~300 time steps later than the corresponding simulations with central individuals). While the central and majority rules resulted in similar times until the end of the simulation, the later departures by the groups using the central rule were compensated for by shorter travel times, thus groups using the central rule were exposed to risk between the refuge and foraging sites for less time. Compared to the majority rule, these shorter travel times for the average individual can be explained by the groups using the central rule being more cohesive, with shorter latencies between the first and other individuals when departing from the refuge.

Increasing $\omega_{difference}$ (the difference between individuals in their tendency to move toward the foraging site, S4 Fig), $p_{baseline}$ (the default level of boldness of individuals in each group, S5 Fig), $d_{food}$ (the distance the group needed to travel, S6 Fig), and the group size $n$ (S7 Fig) in general had little effect on the differences between the social rules. However, the benefit of having social interactions in reducing the time until all individuals had arrived at the foraging site increased as $\omega_{difference}$, $d_{food}$ and group size increased and as $p_{baseline}$ decreased. These trends can be explained by social interactions allowing bolder individuals to have social influence on the slowest individual in the group (which determines the time until the end of the simulation) by reducing the time taken by the slowest individual. $\omega_{difference}$ and group size increase the boldness of the bolder individuals in the group and if social interactions are present, their influence reduces the time taken by the slowest individual. In contrast, the slowest individual does not become bolder and hence faster as these parameters increase when there are no social interactions (Figs 3A–3C and S3A–S3C). This also explains why increasing the boldness of the shyest individual (by increasing $p_{baseline}$) reduces the difference in the time taken between the social and non-social rules, as the time to the end of the simulation for groups with social interactions are much less affected by $p_{baseline}$ than groups without social interactions (S1 Fig panel A). Increasing $d_{food}$ slows the time taken until the end of the simulation to a greater extent in non-social groups, thus as $d_{food}$ increases, the benefit of social interactions increases as social

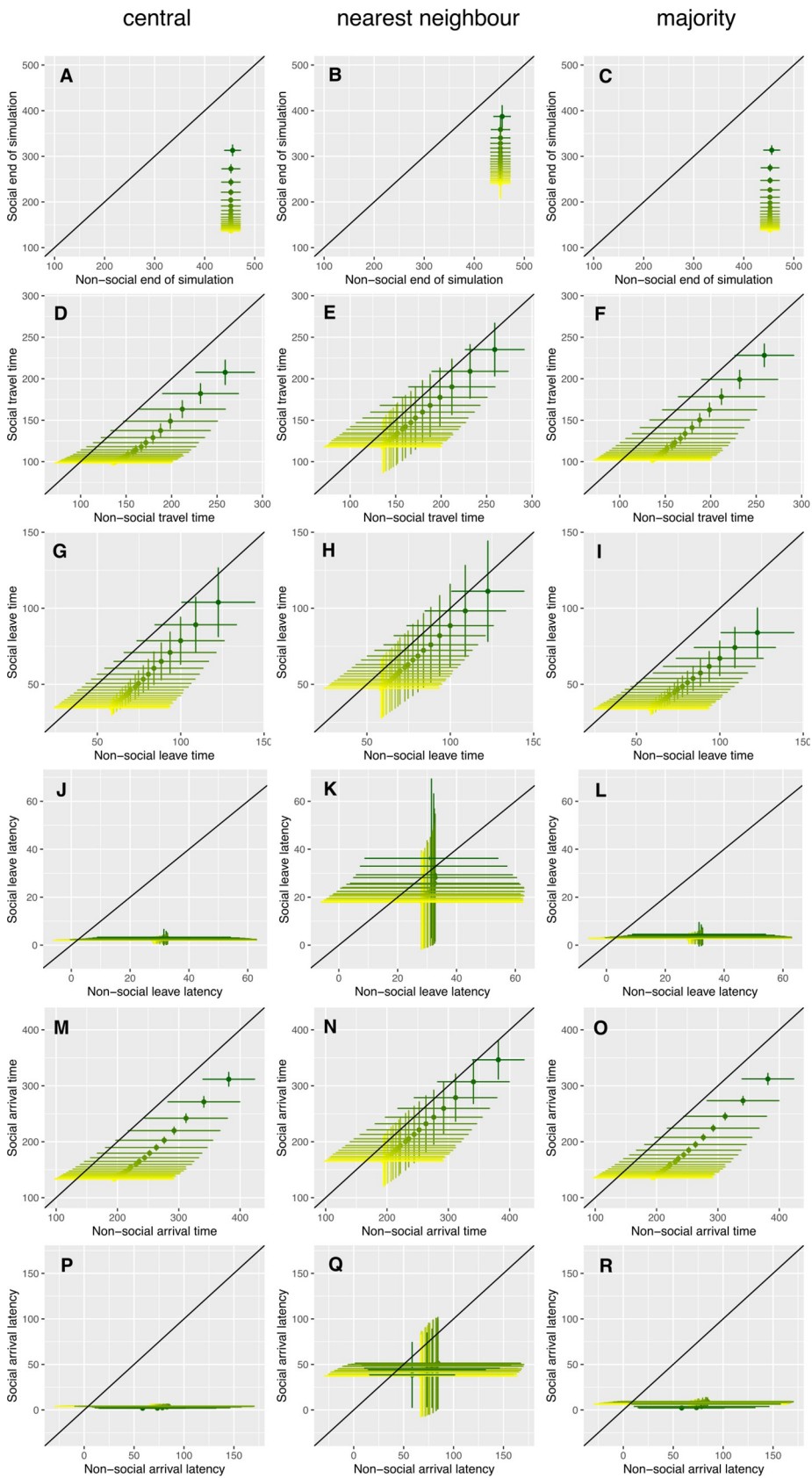

**Fig 3. The effects of the outward probability adjustment parameter $\omega_{difference}$ on timing of measures of group behaviour.** This figure compares the three social interaction rules with simulations where all individuals behaved non-socially. Within each panel, $\omega_{difference}$ is systematically increased from 0.0001 (dark green) to 0.0020 (yellow), with the colour gradient representing equally-sized increments of 0.0001 between these values, and other parameters are set as described in the 'model exploration' methods. Each panel plots the results of a social rule against the non-social values, and the non-social results are therefore identical in the panels for the three different social rules, and are replicated here to allow comparison between social and non-social behaviour. The black diagonal line on each panel represents the scenario where social and non-social individuals are behaving identically; if points fall below this line, the measured behaviour is happening faster with the social rule; if points fall above the line, the non-social rule causes the behaviour to happen faster. The panels show **A**)–**C**) the mean time until the end of a simulation, indicated by the last individual moving to the foraging site that is $d_{food}$ units away from the refuge; **D**)–**F**) the mean time an individual spends travelling between the threshold distance close to the refuge and the foraging site; **G**)–**I**) the mean time that an individual first passes beyond the threshold distance away from the refuge; **J**)–**L**) the mean latency shown by other group members to pass the threshold distance once the first individual has left; **M**)–**O**) the mean time at which the first individual arrives at the foraging site beyond $d_{food}$; **P**)–**R**) the mean latency shown by other group members to arrive at the foraging site once the first individual has arrived. All datapoints show the mean value (± SD) for 10,000 independent group simulations.

influence acts a buffer on the speed of the slowest individual in the group. S4–S7 Figs also demonstrate that there are large variances around these results, and the different rules tend to overlap in their performance despite the obvious qualitative differences, suggesting that there is a lot of noise (from stochasticity within both individual behaviour and between simulations) in the behaviours recorded.

Thinking forward to empirical tests of the model predictions, it is important to quantify the amount of variation between runs of the simulations, as the overlap between the results generated for differing values of the same parameter dictates how easy it would be to successfully identify a true difference between experimentally manipulated groups. For example, in Fig 3A, the vertical error bars (denoting the standard deviation of the measured end of simulations for groups using the central social behaviour, with the change in line colour representing a systematic increase in $\omega_{difference}$) show very little overlap. Conversely, looking at the horizontal (non-social) error bars in the same panel, there is nearly complete overlap for all values of $\omega_{difference}$, suggesting that it would be impossible to discern a difference regardless of the value of $\omega_{difference}$ used in a simulation. Higher values in Fig 4 denote cases where most variation in a measured behaviour occurs between different values of the parameter rather than the variability being between replicates with the same parameter values. This suggests behaviours that would be useful to measure in empirical studies either manipulating these parameters (*e.g.* by manipulating group composition to vary $\omega_{difference}$) or relying on naturally occurring differences in these parameters, such as comparisons between populations under different selection pressures. Low values denote cases where the amount of variation within responses to a given value of a parameter are similar to that seen from all values of the parameter. In general, the time until the end of the simulation, the travel time and the arrival time were sensitive measures allowing the effects of the manipulated parameters to be detected, although for the time to the end of the simulation in particular, this did depend on the social interaction rule being used (note the log scale). The leave time, leave latency and arrival latency did not perform as well, especially when testing for the effects of differing distances to the foraging site (also see S2 Fig panels C-F).

Groups do not consist of individuals behaving in exactly the same way: the $\omega_{difference}$ parameter initialises the differences in movement probability between a group's members from the very start of the simulation. This imposed difference means that some individuals are behaviourally predisposed to leave the refuge and move to the foraging site earlier in the simulation than others. If we consider a group where its members are using the non-social behavioural rule, it is equivalent to the group's members moving on their own, without social interactions.

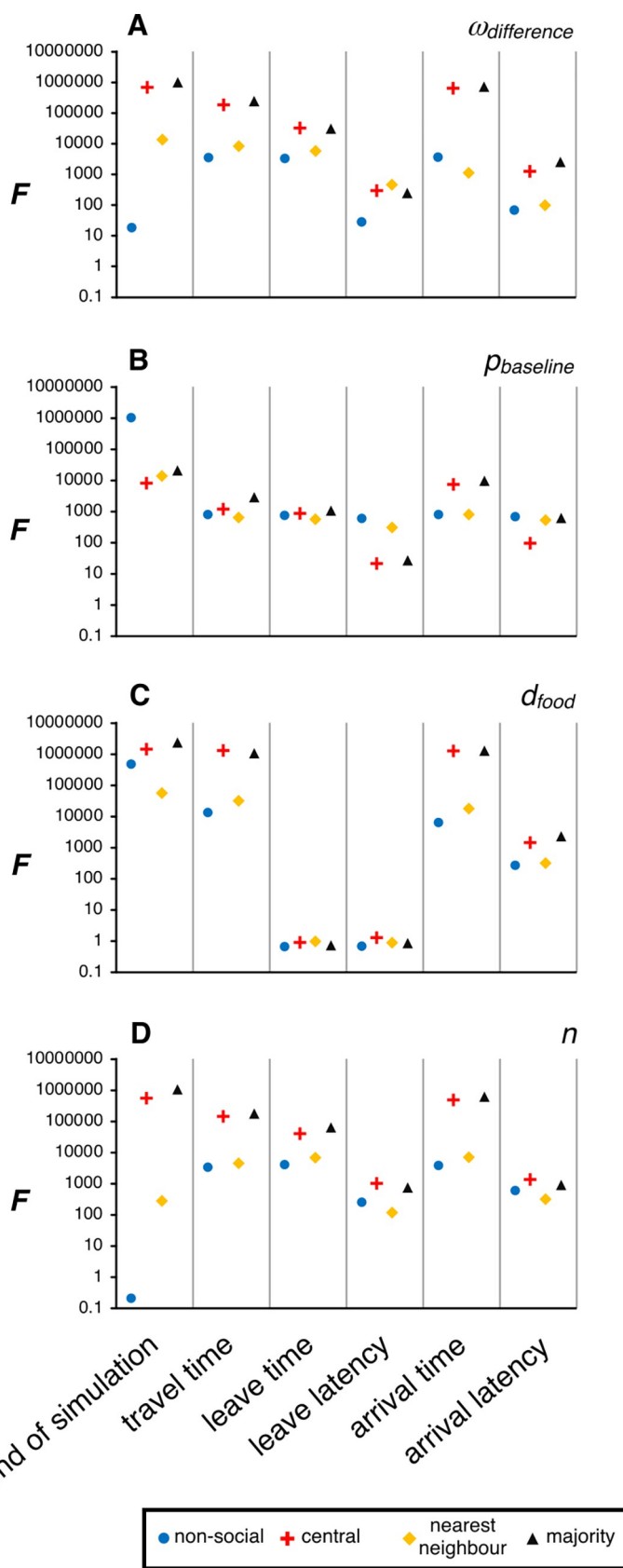

**Fig 4. Quantifying variability within and between parameter sets.** *F* values describing the ratio of within- and between-parameter-set variance for the mean simulation value of the six behavioural measures that were recorded, when the parameter being systematically altered was: **A)** probability adjustment parameter, $\omega_{difference}$; **B)** baseline time-dependent probability of moving outward, $p_{baseline}$; **C)** distance to the foraging site, $d_{food}$; or **D)** group size, *n*. The shape and colour of the points indicates the social interaction rule that was followed within a simulation set.

If we compare these non-social results to the behaviour of identical groups following social rules in an identical environment, we can explore the effects of the social rules on group members. Fig 5 shows the (rescaled) movement times for individuals within a simulation. Individuals using the non-social rule behave as would be expected: those that were behaviourally predisposed to leave the refuge and move earlier (*i.e.* bold individuals) took less time to travel to their destination (as shown by travel times in Fig 5A falling as the individual's identity–corresponding to the probability of it moving outwards–increases), with both their leave times (Fig 5E) and arrival times (Fig 5I) correspondingly falling. Adding social rules did not completely remove the effect of $\omega_{difference}$ on the order in which individuals arrived, but did make it much more difficult to discern between individuals. This is shown by the greater overlap in the distributions of times for all individuals within a simulation, especially between the 'shy' individuals least likely to start moving towards the foraging site (the lower numbered individuals in Figs 5B–5D, 5F–5H and 5J–5L). Table 2 quantifies this overlap in behavioural metrics for individuals, showing much lower overlap (higher *F* values) for the non-social rule. The behaviour being measured was also important; it would be very difficult to discern between individuals if using leaving time as a metric (as can be seen in Fig 5D and the corresponding low *F* value in Table 2). Arrival time gives the best opportunity for discerning between individuals (the high *F* value in Table 2), although Fig 5F demonstrates that this difference is probably going to be more visible in the individuals that are predisposed to moving earlier (those individuals with higher values in Fig 5). For the travel and leave times, individuals were easier to discern if they were using the nearest neighbour interaction rule compared to the other social interaction rules, although they were easier to discern when using the majority rule when arrival times were measured. Table 2 demonstrates that these trends are similar when systematically altering $p_{baseline}$ and $d_{food}$.

Differences between the behaviours shown by specific individuals within a group are most visible in those individuals that are most likely to move towards the foraging site, as can be seen in individual 10 in the social behaviours shown in Fig 5: the behaviour of these individuals appears to be least influenced by other members of the group, with a smaller range of variation across the simulations showing that these individuals still tended to be last to conduct behaviours when they were moving within a socially-behaving group. If we look at how the behaviour of individuals changes between using the non-social rule and one of the social rules (Fig 6), it is apparent in many cases that individual 10 shows little change in behaviour, with behavioural differences close to zero meaning that there is no difference when using non-social or social rules. This is not the case when considering the leave time (Fig 6D and 6F, and Fig 6E to a lesser extent), where a positive value indicates that even the most motivated individuals tend to take longer to move away from the start point when behaving socially compared to when they behave independently of other individuals. Most of the other individuals show this slower behaviour in travel time, leave time and arrival time when social, with the exception of individual 1 (the least likely individual to move away from the start point when behaving independently), which moves faster when behaving socially than when non-social.

For individuals who are neither the fastest or slowest to move to the foraging site when behaving independently (*i.e.* those with intermediate values of $\omega$), there were moderate ordering changes when comparing social and non-social behaviours, with individuals tending to

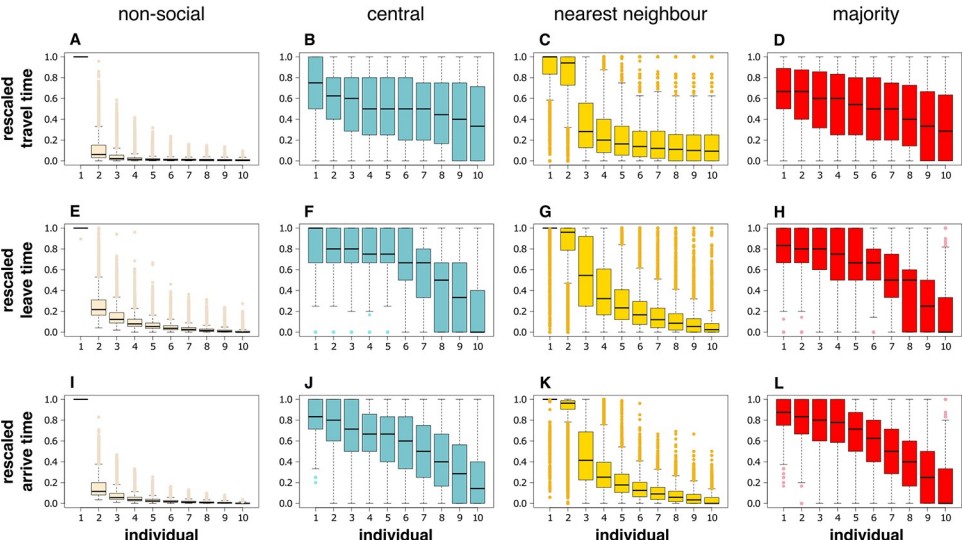

**Fig 5. Illustrative examples of mean rescaled individual behaviour.** Panels compare non-social individual behaviour and the three social behaviours. Individuals are labelled 1–10, where 1 is the individual with the baseline levels of all parameters, and incremental increases in the label represent the incremental increase by $\omega_{difference}$ as described in the methods. This means that when all individuals within a simulation are behaving independently (non-socially), individual 1 is the least likely to start moving towards the feeding site (the shyest), and individual 10 is the most likely (the boldest). Panels **A-D** show rescaled travel times, where 0 is the travel time of the individual who reaches the feeding site in the shortest time, and 1 is for the individual who takes the longest; **E-H** show rescaled leave times (at which an individual first crosses a threshold 10 units away from the start point), and **I-L** show rescaled arrival times at the feeding site. **A**, **E** and **I** show the behavioural metrics for individuals behaving non-socially (independently) within a simulation; **B**, **F** and **J** show the metrics for individuals using the central social rule; **C**, **G** and **K** show the metrics for individuals using the nearest neighbour rule; and **D**, **H** and **L** show the metrics for individuals using the majority rule. All boxplots show the median and interquartile values of the mean rescaled behavioural metric, and the tails show $1.5 \times$ interquartile range, with points representing outliers. Data shown for dataset where $\omega_{difference}$ has been systematically altered in order to explore variation in response to this parameter (200 replicates each for $\omega_{difference}$ = (0.0001, 0.0002, . . . 0.0020), pooling all of these simulations together for each of the figures), with $p_{baseline}$ = 0.001, $d_{food}$ = 100 units and $n$ = 10.

**Table 2.** *F values describing the contributions of both the model parameter being varied and the 'boldness' of the individual within a simulation, according to the behavioural metric measured and the parameter being varied.*

| | Non-Social | | Central | | Nearest Neighbour | | Majority | |
|---|---|---|---|---|---|---|---|---|
| | Parameter $F$ | Individual $F$ | Parameter $F$ | Individual $F$ | Parameter $F$ | Individual $F$ | Parameter $F$ | Individual $F$ |
| $\omega_{difference}$ | | | | | | | | |
| travel time | 4797 | 157054 | 15 | 291 | 51 | 4876 | 48 | 318 |
| leave time | 911 | 133875 | 2 | 2526 | 23 | 8487 | 6 | 2956 |
| arrival time | 8021 | 200045 | 41 | 1966 | 10 | 19566 | 59 | 3753 |
| $p_{baseline}$ | | | | | | | | |
| travel time | 583 | 97981 | 3 | 2311 | 16 | 9679 | 6 | 7808 |
| leave time | 251 | 30072 | <1 | 447 | 8 | 1903 | 1 | 316 |
| arrival time | 1706 | 157742 | 6 | 11168 | 22 | 10625 | 13 | 34995 |
| $d_{food}$ | | | | | | | | |
| travel time | 141 | 77372 | 18 | 1156 | 22 | 6406 | 44 | 3253 |
| leave time | 1 | 37140 | 1 | 446 | 1 | 1757 | 2 | 313 |
| arrival time | 157 | 326247 | 34 | 4843 | 30 | 8211 | 52 | 11270 |

Higher values in the 'Individual $F$' mean that it is easier to identify a specific individual by measuring its behaviour alone.

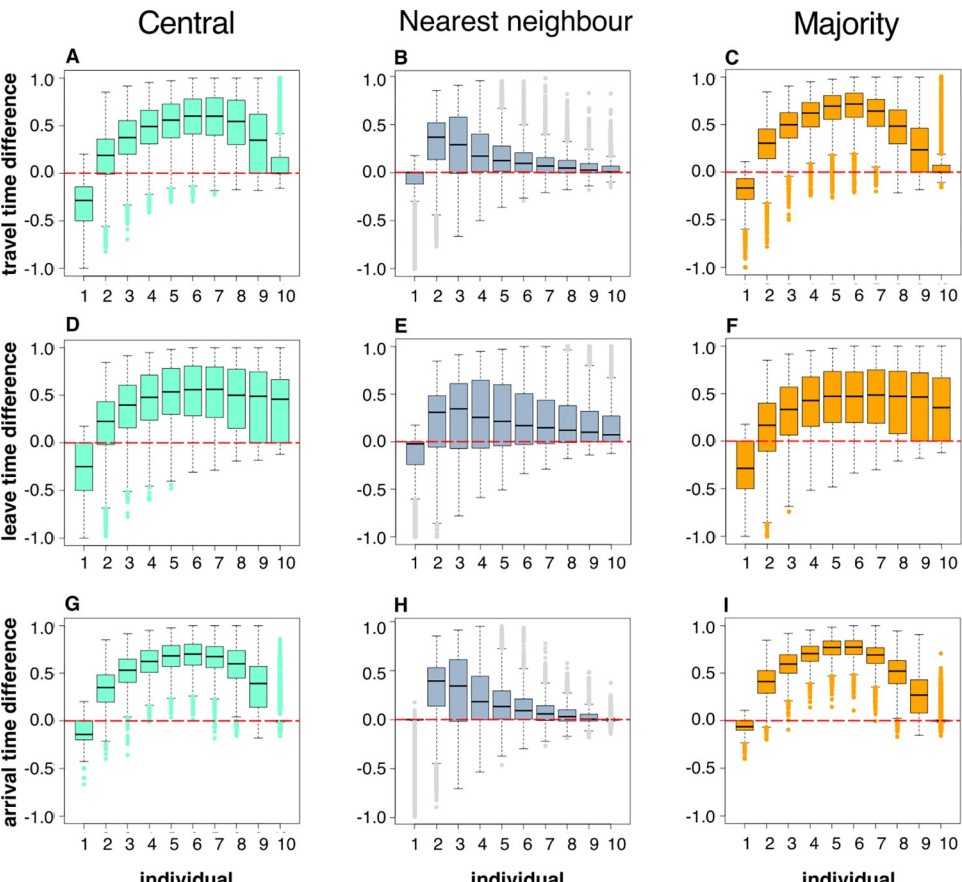

**Fig 6. Mean rescaled differences in behavioural metrics between the non-social rule and each of the social interaction rules for each individual.** Positive values mean that individuals switch the observed behaviour to later in the hierarchical group order when behaving socially, and negative values mean they switch to earlier in the group order when behaving socially. This means a value of 1 represents the case where the individual is always first to finish/leave/ arrive when behaving non-socially, and always the last to finish/leave arrive when behaving socially, a value of -1 represents the case where the individual is always last to finish/leave/arrive when non-social and always first to finish/ leave/arrive when social, and a value of 0 represents the case where the individual does not change the ordering of its behaviour between social and non-social behaviours (*e.g.* the third individual to arrive when performing a non-social behaviour is also the third individual to arrive when performing a social behaviour). **A**, **B** and **C** show the difference in travel time; **D**, **E** and **F** show the difference in leave time; and **G**, **H** and **I** show the difference in arrival time. **A**, **D** and **G** give the values when comparing the central and non-social rules; **B**, **E** and **H**, the nearest neighbour and non-social rules; and **C**, **F** and **I**, the majority and non-social rules. See the legend of Fig 5 for a description of the boxplot ranges and the parameter set used for generating the figure.

shift an observed behaviour to later in the hierarchy when behaving socially. This difference is greatest for individuals in the 'middle' of the group movement parameter set for central and majority social rules (*e.g.* Fig 6A), or greatest for the second-most 'shy' individual (individual 2) for the nearest neighbour rule (*e.g.* Fig 6B). Contrary to the other nine members of the group, the most 'shy' individual (individual 1) tended to shift its ordering to earlier in the hierarchy when behaving socially. This reflects the cohesive nature of the social group behaviours, where the individual that would normally take longest to travel to safety is caught up within the group and consequently has an opportunity to conduct a measured behaviour before at least one other of its colleagues, dragging the values shown in Fig 6 downwards from the zero values. Because Fig 6 shows standardised values, caution should be taken to not interpret the trends shown as indications of behaviours becoming faster or slower. Instead, the broad

message of this figure is that social behaviours mean that group members are more likely to be behaving together, and their individual behavioural identities become less easy to identify when observing the group behaving together.

## Discussion

It is well established in experimental studies that individuals in groups can forage more successfully than solitary individuals [67–69]. Our model replicates this effect in a simple, idealised scenario by comparing simulations of individuals with no social behaviour, *i.e.* where they acted independently, to simulations with social interaction rules. However, speeds were not always faster in social groups. The social rule of moving to the centre of the group actually increased the time taken to leave the refuge, but reduced the time taken to travel to the foraging site. McDonald *et al.* [45] showed a similar trend, where the median time taken to leave the refuge was marginally longer in groups compared to solitary individuals, but the time taken to cross the arena was considerably faster in groups. The later departures from the refuge for groups using the central rule were more than compensated for by faster travel times between the refuge and foraging site, so that these groups arrived sooner than groups using the other rules. The central rule thus reduces exposure to predation risk during travel to a foraging site but still allows earlier access to the foraging site, and hence would be favoured by natural selection.

Our results also demonstrate that simple social behaviours can result in consistent inter-individual differences in behaviour (widely referred to as personality variation) being repressed. This echoes experimental results [45,51], where behaving as part of a group reduces the expression of behaviours that are consistently shown when an individual is on its own. As McDonald *et al.* [45] note, although social conformity suppressing personality variation is well documented [50,70,71], the mechanisms behind this conformity are not known. McDonald *et al.* [45] provide evidence that quorum-like consensus decision-making might be driving this behaviour (similar to the majority-movement rule that we consider), but the model we present here demonstrates that other social mechanisms can result in this effect as all three of the social behaviours considered led to social conformity occurring. It is likely that other social behaviours exist that will give similar behaviours (compare for example the various different rules suggested for selfish herding: [8,57–60]), and careful experimentation would be required to identify whether these are appropriate for explaining the behaviour seen (such as the technique used by [72]). The three social interaction rules we present consider an individual as paying attention to different individuals in the group. Huth and Wissel [5] suggest that paying attention to many neighbours is important–just paying attention to the nearest neighbour cannot realistically drive a group's behaviour. We demonstrate that personality erosion occurs both when individuals are considering a single neighbour (the nearest neighbour social rule) and when individuals are accounting for the behaviour of most or all of the group (the central and majority social rules), although the effect of conformity was stronger when a larger proportion of the group are considered. Not only do our results demonstrate that simple social interactions can make discerning between different personality types in groups difficult, but they also imply that selection pressures that would act on personality variation in non-social animals may have no, or a weakened, effect in social groups where conformity reduces the realised differences between individuals [52].

Our model considers a simple drive to move from shelter to a food source, which we allow to differ between individuals. Personality variation can be based on differences in metabolic rate, where those with greater metabolic requirements will consistently show greater risk-taking behaviour which increases access to food [73], although, conversely, satiation of bolder

individuals after feeding can result in switching of leader-follower roles [74]. Our model can thus represent a simple caricature of a group of individuals being driven by their energetic reserves, where those individuals moving sooner and with a faster increase in their probability to move could be seen as 'hungry' individuals. This is consistent with experiments where food-deprived individuals are mixed with satiated ones, and the hungrier individuals tend to move to food sooner and may drive the behaviour of the rest of the group [36–38,75–77]. Although in our simulations the predetermined requirement to move towards the food is shown by all individuals, the social interaction rules mean that the movement of individuals is determined not just by 'hunger' but also by the behaviour of the other members of the group. Furthermore, our results suggest that the lag shown by normally bold individuals (that is revealed by comparing simulations with social and non-social behaviours) may be driven by the reluctance of less bold individuals to move, mirroring the large influences that individuals with specific personality traits may have on the group's behaviour [78].

We acknowledge here that the one-dimensional nature of our model could have influenced the results that we describe. A one-dimensional system was chosen because it gave the simplest caricature of the stickleback system of McDonald *et al.* [45] that we were attempting to investigate. A one-dimensional system like this is attractive for initial investigation as it has a relatively small parameter space and it is feasible that it could be translated to an analytical model, but adding dimensionality could impact on the behaviours seen, especially as collective behaviour may be strongly impacted by real-world physical constraints [79]. Similarly, we acknowledge that the social behaviours considered (one metric, based on calculating a mean centre of the moving shoal, and two topological, based on proximity and counting) are only three of a wide range of possible rules that could be used, and were arbitrarily chosen here to provide an alternative to the non-social scenario. Extending the model to two or three dimensions could allow us to exploit directional/visual constraints (*e.g.* [7,80–83]) as well as topological and other non-metric rules (*e.g.* [84,85]), and could reveal much more about the impact of social rules and personality on collective decision-making.

Although our model considers a driving force that brings individuals to a food source, we do not explicitly consider energetic reserves or any other physiological processes [86]. Movement and changing direction may be costly (as is considered by [76]). Nor do we consider a cost to associating closely together, where collision risk could be a danger (as is considered in the inner repulsion zone considered in many models of collective behaviour [4–9]). Similarly, we make no assumptions about predation in this model, although the social behaviours we consider echo those considered for social herding behaviour [57–60], and it has been demonstrated that individuals may use more complex social behaviour rules [87], pay attention to more neighbours [88], and make more egalitarian collective decisions [89] when exposed to predation risk.

Given that the rules governing the behaviour of individuals relates to both their internal state and the behaviour of other individuals (where the behaviour of those other individuals is dictated by their own internal state), it would be possible to use modelling tools such as stochastic dynamic games (*e.g.* [32,33]) to calculate optimal social rules, which would mean that we could use a little less guess-work in suggesting the heuristics that individuals are using [90]. Following the experiment presented by McDonald *et al.* [45], we implicitly assume here that all individuals are equally aware of the food source, and alternative assumptions would need to be made to consider the case where only a few individuals are informed [25,91]. Similarly, careful experimental manipulation of the energetic state of individuals in groups may reveal further subtleties in how individuals alter their behaviour to suit their immediate personal condition [92]. Potentially confounding hierarchical differences, or differences in being able to compete for food [2], between individuals would also need to be accommodated [33,93], as

social hierarchies may directly confound the energetic states of interacting individuals [94–97]. Regardless of these additional complexities, we have demonstrated here that simple social behavioural rules can drive conformity behaviour in groups, eroding consistent behavioural differences shown by individual animals.

## Software availability

Annotated *Netlogo* code [64] is freely available on *figshare* at https://doi.org/10.6084/m9.figshare.19391651. Annotated *R code* used for generating the figures [98] is freely available on *figshare* at https://doi.org/10.6084/m9.figshare.21996242.

## Supporting information

**S1 Fig. The effects of the baseline time-dependent probability of moving outward $p_{baseline}$ on timing of measures of group behaviour, comparing the three social rules to simulations where all individuals behaved non-socially.** Within each panel, $p_{baseline}$ is systematically altered increased from 0.0001 (dark green) to 0.0020 (yellow), with the colour gradient representing equally-sized increments of 0.0001 between these values, and other parameters are set as described in the 'model exploration' methods. See legend for Fig 3 for more detail.
(PDF)

**S2 Fig. The effects of the distance of the foraging site $d_{food}$ on timing of measures of group performance, comparing the three social rules with simulations where all individuals behaved non-socially.** Within each panel, $d_{food}$ is systematically altered between 20 (dark green) and 200 distance units (yellow), with the colour gradient representing equally-sized increments of 10 units between these values, and other parameters are set as described in the 'model exploration' methods. See legend for Fig 3 for more detail.
(PDF)

**S3 Fig. The effects of the group size $n$ on timing of measures of group performance, comparing the three social rules with simulations where all individuals behaved non-socially.** Within each panel, $n$ is systematically altered between 5 (dark green) and 50 individuals (yellow), with the colour gradient representing equally-sized increments of 5 individuals between these values, and other parameters are set as described in the 'model exploration' methods. See legend for Fig 3 for more detail.
(PDF)

**S4 Fig. The effects of the outward probability adjustment parameter $\omega_{difference}$ on group performance when the social interaction rule is varied.** The panels show summaries of the difference in the behavioural metrics (the labels at the left of the figure describe each statistic) when comparing the six possible pairs of social behaviour (indicated by the labels at the top of the figure), noting that the differences were calculated by comparing the behavioural metrics for simulations of groups experiencing the same starting conditions but following different behavioural rules. Boxplots represent the median and quartile values for each simulation, with whiskers representing the minimum of the maximum value and 1.5 × interquartile range, with points representing outliers beyond this range.
(PDF)

**S5 Fig. The effects of the baseline outward movement probability $p_{baseline}$ on group performance when conducting different social behaviours.** See the legend to S4 Fig for details.
(PDF)

**S6 Fig. The effects of the distance to the foraging site $d_{food}$ on group performance when conducting different social behaviours.** See the legend to S4 Fig for details.
(PDF)

**S7 Fig. The effects of group size $n$ on group performance when conducting different social behaviours.** See the legend to S4 Fig for details.
(PDF)

## Author Contributions

**Conceptualization:** Sean A. Rands, Christos C. Ioannou.

**Data curation:** Sean A. Rands.

**Formal analysis:** Sean A. Rands.

**Funding acquisition:** Sean A. Rands, Christos C. Ioannou.

**Investigation:** Sean A. Rands.

**Methodology:** Sean A. Rands.

**Software:** Sean A. Rands.

**Visualization:** Sean A. Rands, Christos C. Ioannou.

**Writing – original draft:** Sean A. Rands.

**Writing – review & editing:** Sean A. Rands, Christos C. Ioannou.

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
