## [Decision Letter · Decision Letter 0]

10 Jul 2022

Dear Dr. Rands,

Thank you very much for submitting your manuscript "Personality variation is eroded by simple social behaviours in collective foragers" for consideration at PLOS Computational Biology.

As with all papers reviewed by the journal, your manuscript was reviewed by members of the editorial board and by several independent reviewers. In light of the reviews (below this email), we would like to invite the resubmission of a significantly-revised version that takes into account the reviewers' comments. All the Reviewers raised important points about the underlying assumptions of your model, and I would like to encourage you to pay special attention to them.

We cannot make any decision about publication until we have seen the revised manuscript and your response to the reviewers' comments. Your revised manuscript is also likely to be sent to reviewers for further evaluation.

Sincerely,

Ricardo Martinez-Garcia

Associate Editor

PLOS Computational Biology

Natalia Komarova

Deputy Editor

PLOS Computational Biology

Reviewer's Responses to Questions

**Comments to the Authors:**

Reviewer #1: The manuscript Personality variation is eroded by simple social behaviours in collective foragers authored by Rans and Ioannou and submitted to PLOS Computational Biology reports results regarding an theoretical study exploring the interaction between personality and social behaviour in predicting the efficiency of foraging. The model chosen is a simple 1-dimension (1D) depiction of agents moving from a refuge towards a food source or other group members, depending on their personality (a range of boldness) and social behaviour (three distinct rules of social interactions are tested).

Overall, I found the study well executed, the research question interesting and topical for the research field, the analysis thorough and the results convincing. The Introduction and Discussion sections put the study in context well. I only have a few minor comments.

The first one concerns the lack of justification of choosing a model spatial in only one dimension (agents move on a line). It might be worth for the manuscript to bring some justifications regarding this point and whether or not the authors think this choice would affect how general their results are. Indeed, in several systems, going from 1D to 2D change results in some instrumental ways (e.g. the Ising model). Most of the empirical systems referred to in the manuscript depict experiments in fish, often recorded in 2D and occurring in 3D, which does not make the 1D choice obvious.

The second (minor) comment regards the difficulty to follow the analysis in several sections/paragraphs of the manuscript. Given the number of subfigures in the manuscript figures, it is, I think, important to precisely refer to specific subfigures in the Results section, for the reader to follow the authors’ train of thoughts. I have tried to list below some instances where, I think, wording or referred subfigures were unclear (to me at least).

I would conclude my main, yet minor, comments by suggesting that the choice of placing the current manuscript within the personality literature might lower a bit the generality of the predictions of the study. It seems to me that the parameter w (controlling for different tendencies in the group to initiate a movement towards the food source), which is used here to depict the boldness of group members, could also depict the end result of other effects changing the tendency of individuals to move towards a food source. For instance, this parameter may also depict the private information of individuals regarding the current quality / quantity level of a known food source. Or, as mentioned in the Discussion (paragraph starting L491) depict the internal state of individuals (e.g. how hungry individuals are). So it seems the current model could describe a variety of effects affecting the internal/cognitive state of agents. Therefore, restricting the article to investigate personality rather than simply the varying tendency of group members to move towards a food source / to be temporary leaders, could be seen as reducing the scope of the model. On a side note, it could be important to add L144/145 that w depicts boldness only if individuals have all the same prior knowledge regarding the location of the food source. A bold personality itself is not sufficient to drive agents towards the food source, only to get out of the refuge, without directionality. This clarification is only made (I think) L531.

Please find below a list of minor comments and possible typos:

L124 boldness if the individual > boldness of the individual

L142 To improve clarity, this inline equation, could be after L124, as a labelled equation

L156 the description of the ‘central’ rule is not clear to me. The text reports two sums (weights, L166). I do not understand what the authors mean there. From the equation it seems there is one sum for each individual rather than one sum per possible direction of movement, as suggested by the sentence the way I understand it.

L293-299: I do not see the exceptions of social interactions leading to faster/sooner behaviours on the Figure 1 g-i (subfigures which are not clearly referred to in this sentence), which I must not read properly. In these three subplots, individuals behaving socially have on average a shorter ‘leave time’ than the non-social ones, in agreement with L291, and therefore in contrast to a sentence starting by ‘Exceptions to this’. I also do not see that ‘central’ and ‘majority’ rules are very different and that ‘nearest neighbour’ is an intermediate – to me the latter is a bit different from the other two if we assess the figure from the standpoint of the effect of social rules against non-social rule? I do see that individuals in the ‘majority’ rule leave faster than in the other two treatments. Maybe this paragraph could benefit from some clarifications and rewording.

L301: it would be easier to read if Figures were referred to as Figure 1 j-l rather than row ‘d’ which is not mentioned on the figure.

L308: maybe precise which figure we should look at

L320-323: to make the sentence clearer and easier to navigate through figures, precise that we are looking at the row ‘difference between ends of simulations’ in the first three columns of Figs 4-7.

L329: Fig S3a I assume. Should Figures 1 and S3, b and c also be referred to here?

L408: the smallest or a smaller

L405-410: the wording makes in my opinion the sentence unclear – by starting referring to the individuals most likely to move towards the foraging site (i.e. individuals #10) and then referring to an example taken with individuals #1. I think that going the other way around or making the connection between the first part of the sentence to the second more explicit in the paragraph would improve clarity of the message.

L480: just paying attention TO the nearest neighbour?

Reviewer #2: This is an interesting study that presents results from models comparing how individual differences in moving agents diminish when social interaction rules are altered. Although I am not a modelling expert, the line of argumentation in the manuscript is solid and I like how the authors introduce and discuss their findings.

Points of critique

As for every modeling approach a lot of assumptions have to be made and I would like to have the authors explain in more detail why they decided to use their 3 social rules (center, NN and majority). This is simply because empirical tests in birds and fish and also other modelling approaches showed that interactions in collectives often follow rather sophisticated rules different from purely metric (Voronoi Interactions, see Gautrais et al. 2012; https://journals.plos.org/ploscompbiol/article?id=10.1371/journal.pcbi.1002678; see Ballerini et al. 2008 https://www.pnas.org/doi/full/10.1073/pnas.0711437105, available visual fields: Rosenthal et al. 2015 https://www.pnas.org/doi/10.1073/pnas.1420068112; Bastien and Romanczuk 2020 https://www.science.org/doi/full/10.1126/sciadv.aay0792; Bierbach et al. 2020 https://peerj.com/articles/8974/ )

So, giving that animal groups may not (just) use the 3 herein examined social rules (maybe they don’t use any of them?) but may rely more on sensory inputs and topological rules, how can the current results be generalized then?

While used assumptions of the model have to be justified better, the overall aim of the study (= to model consistent behavioral differences of agents in groups) is surely highly interesting and not much has been done into this direction on the modelling side (but see also Jolles et al 2017 https://www.cell.com/current-biology/fulltext/S0960-9822(17)31013-8 .

Reviewer #3: Dear editor and authors,

Thank you for sharing this manuscript with me, and I apologize for the delay in turning it around. It’s been a wild few weeks here, full of surprises.

The manuscript takes aim at an important question, namely how we balance a rich understanding of inter-individual differences with the cohesion and consistency we observe within many types of collectives. I think this is an important question and one that has received increased attention over the last several years, for good reason. They approach this theoretically, using a model of a quite simple/straightforward context in which a heterogenous collective is moving between refuges and foraging. There is much to like about this paper, I quite enjoyed reading the introduction and am thrilled to see full, functional code supplied alongside the manuscript.

Where I get hung up, is in the work it took me to make sense of what the model was doing from the text alone, and (given that) what the results/figures were telling me. Some of this could be addressed in the writing, for instance in the paragraph starting in L139. “If the refuge site lies between itself and the foraging site, it moves towards the foraging site”. Is it not moving towards the refuge site too? I don’t totally understand the purpose of this rule, is it to avoid some sort of artifact where the group travels away from the foraging site?

Next, if it is between the two, it either forages or behaves socially. This was clear after a bit, but none of these rules bring the agent towards the refuge site so I’m still left wondering why it exists as a construct here. “starting location (for instance a refuge)” might be more clear. After working through the general form of the model, I then found myself staring at a large number of parameters---some explored, some fixed---and having sufficient trouble juggling them all to make full sense of whether the results explored in the paper are generalizable or some quirk of parameter space. I highly recommend a table/figure/diagram to help the reader along. Although, perhaps uniquely I’m a bit slower here… taking longer to move from the refuge of not understanding to foraging on the model.

In my mind, there seems to be a parallel model of this that would be analytically tractable, perhaps borrowing from the engineering literature on expected failure times with cascading/coupled failure. I imagine it would produce much the same result, with variation in times to failure across parts being less important if the parts are substantially coupled.

I’d like to see this published, but perhaps not in its current form. Here’s what I would suggest: First, try to consolidate parameters wherever possible. I’m not quite sure if this is doable, but it would greatly help with model interpretation. Next, please add a table of the parameters, their values explored, and their meaning. A figure/diagram of the model would be super helpful. In the results, I would consider thinking of other summary statistics that may be a bit more intuitive, as I found the ANOVA-derived presentation a bit hard to ground back to the phenomenon at hand. It would be nice, where possible, to frame the model evaluation in something more clearly biologically relevant (time to forage, variation in time to forage, etc..)

On “Personality”

In discussions of personality, particularly in fish schools, it strikes me that simply swimming faster (or moving more) could be mistaken for boldness. I do quite like that the authors don’t fully hang their hat on personality being innate and note that it may well be things such as differences in morphology or transient differences in behavior.

The authors’ acknowledgment of this makes me wonder (somewhat) whether the use of the phrase personality was to appease potential reviewers who prefer that term to more generic ones (inter-individual differences, heterogeneity, etc..). From the perspective of the model, is the term personality necessary, or is the model asking a much more general question? I won’t hold up the paper on this, but It may be worth considering whether broader language would better reflect the mode. Something along the lines of “interindividual differences, such as personality, swim speed body size, condition, etc…”

Smaller points:

L30: I quite dislike novelty/priority claims, are we certain this is the *first*? It’s a big literature.

L131: Is there a period oddly placed in the middle of the formula?

L142: if i goes from 1-n (L128) then w_i is integer-valued and ranges from 0 to N-1 (L131). If you’re adding w_i to p_{i,t} how do you recover a probability? I see that there is w_difference…. Is that just a scaler factor that is something like 1/N? If so, can’t adding that to each time step cause the probability to be greater than 1?

L195: why these values for baseline? Does it matter? It’s ok if it’s arbitrary, I’m just trying to follow.

L221-233: I don’t really understand why you’d calculate F in an ANOVA to compare things here. Why not just plot some summaries of model outputs? This seems like a strange choice

**Have the authors made all data and (if applicable) computational code underlying the findings in their manuscript fully available?**

Reviewer #1: Yes

Reviewer #2: Yes

Reviewer #3: Yes

PLOS authors have the option to publish the peer review history of their article (what does this mean?). If published, this will include your full peer review and any attached files.

Reviewer #1: No

Reviewer #2: No

Reviewer #3: No
---

## [Decision Letter · Decision Letter 1]

31 Jan 2023

Dear Dr. Rands,

We are pleased to inform you that your manuscript 'Personality variation is eroded by simple social behaviours in collective foragers' has been provisionally accepted for publication in PLOS Computational Biology.

Best regards,

Ricardo Martinez-Garcia

Academic Editor

PLOS Computational Biology

Natalia Komarova

Section Editor

PLOS Computational Biology

Reviewer's Responses to Questions

**Comments to the Authors:**

Reviewer #1: The new version submitted by the authors is a significant improvement on the previous version, with clearer explanations (thanks to new figures describing the simulated set-up and social rules) and descriptions. The authors have addressed all my former comments and suggestions in a sensible way.

**Have the authors made all data and (if applicable) computational code underlying the findings in their manuscript fully available?**

Reviewer #1: Yes

PLOS authors have the option to publish the peer review history of their article (what does this mean?). If published, this will include your full peer review and any attached files.

Reviewer #1: No

---

## [Editor Report · Acceptance letter]

8 Feb 2023

PCOMPBIOL-D-22-00512R1 

Personality variation is eroded by simple social behaviours in collective foragers

Dear Dr Rands,

I am pleased to inform you that your manuscript has been formally accepted for publication in PLOS Computational Biology. Your manuscript is now with our production department and you will be notified of the publication date in due course.

With kind regards,

Bernadett Koltai
